# Job Design to Extend Working Time: Work Characteristics to Enable Sustainable Employment of Older Employees in Different Job Types

Hiske den Boer [1,*], Tinka van Vuuren [1,2] and Jeroen de Jong [3]

1   Faculty of Management, Open University of The Netherlands, 6419 AT Heerlen, The Netherlands;
    Tinka.vanVuuren@ou.nl or tinka.van.vuuren@loyalis.nl
2   Loyalis Knowledge & Consult B.V., 6411 CR Heerlen, The Netherlands
3   Institute for Management Research, Nijmegen School of Management, Radboud University Nijmegen,
    6525 AJ Nijmegen, The Netherlands; j.dejong@fm.ru.nl
*   Correspondence: hiske.denboer@ou.nl; Tel.: +36-30-7326332

**Abstract:** Due to an aging workforce and an increasing structural labor shortage across Western economies, it is important to design jobs for older workers that support their continued employability. The aim of this qualitative study was to investigate how job type (operational, professional and managerial jobs) influences work characteristics older workers need to continue working. Semistructured interviews were conducted with 21 older (55+) Dutch employees working in the health and education sector. A full thematic analysis of interview transcripts was performed, and work characteristics were identified, coded, categorized and compared to discover patterns of similarities and differences between job types. The results show that job types have a number of work characteristics in common: operational job types share autonomy with managers and client interaction with professionals, and professionals and managers share mentorship. Unique work characteristics for operational roles are supervisor support and comfortable workspace. Professionals especially want to use their expertise and flexible working hours, and managers are different because they value personal development and contact with colleagues. In conclusion, the results show that certain work characteristics have a different impact on the design of future jobs for older workers, depending on the type of job of the employee.

**Keywords:** job design; work characteristics; job types; sustainable employability; older workers; extend working time; qualitative research

## 1. Introduction

Many Western countries have been facing the effects of ageing workforces [1]. In the Netherlands the number of people in work is decreasing due to the aging of the total population [1]. Due to these developments, social costs are expected to increase, underlining the importance of involving, recruiting and retaining as many older people as possible in the labor process [1]. To meet the challenges of employability for older workers, age-driven human resources policies initially focused on alleviating the workload for older workers, such as shorter or more flexible working times, while currently policies are more focused on keeping older workers motivated and vital [2], because sustainable employability of employees can be improved by increasing their vitality and work ability [3]. It is increasingly relevant to consider the role that job design plays in finding ways for older workers to stay satisfied and continue working [4]. In particular, job design researchers pointed to the importance of understanding the role of individual differences in examining the effects of work characteristics on work outcomes including satisfaction and motivation [5]. This means that work characteristics will not be experienced in the same way by all employees. Given the diversity of older employees in the current workplace, categorizing work into

operational, professional and managerial roles provides relevant insights into how older workers' jobs should be designed and tailored to these job type differences to encourage them to continue working, now and in the future. Therefore, this qualitative study focuses on how older workers in different job types would organize their future job and what specific work characteristics can motivate them to work longer.

### 1.1. Working Longer and Job Design

Due to rising life expectancy, an economic crisis, inadequate pension systems and a shortage of younger people entering the labor market, older workers are contemplating retiring or continuing to work. Organizations are willing to develop jobs that enable employees, who are part of the aging workforce, to continue to work successfully [4].

There are a number of reasons why organizations want to attract and keep older workers. First, retaining older workers can help companies avoid talent shortages. Labor market economists expect shortages in specific occupations in growth and highly specialized industries, such as engineering, technology and health. These companies anticipate that the loss of talent, irrespective of age, will be a problem and they want to retain older workers in highly professional jobs [1]. Second, older employees are skilled, experienced, and maintain a company's knowledge and relationship networks. When older workers retire, the loss of their expertise and experience can negatively affect the performance of the organization [6]. Finally, older workers have a strong work ethic, take the job seriously and want to be recognized for their work; they are likely to have a more positive attitude toward their work than younger employees [7].

Not only is it crucial for employers to keep older workers in employment, there are also good reasons for older workers to continue working. There are combinations of four main reasons for older workers to work: social (interaction with others), personal (intrinsic rewards), financial (monetary rewards) and generative (knowledge transfer) [8]. Research suggests that older workers are likely to postpone their retirement, because for many employees, staying in the workforce for longer is both inevitable and desirable, not only as a source of income, but mainly because work provides people with a meaningful interpretation of their daily lives [9]. Even though older workers may need income, they place a special premium on feeling engaged and doing meaningful work and, equally important, feeling socially connected to colleagues [6].

While there are forces that encourage older workers to work longer, there are also obstacles that limit their choices. Unemployment in the Netherlands was historically low in 2019, but unemployment only rose among people aged 65 and older. This is explained by the fact that the state pension age has been raised (66 years and 4 months in 2021), as a result of which older workers will participate longer in the labor process. However, if they become unemployed, they will find it more difficult to find a job and this will lead to higher long-term unemployment among older workers [10]. Employment contracts are also subject to change. If older employees work under a collective labor agreement or an individual contract stating that the employment contract will be terminated upon reaching the state pension age, continuing work is only possible if the employer wishes to enter into a new contract [10]. Moreover, older workers are no longer insured against unemployment and incapacity for work, but can be entitled to sickness benefits. People aged 55–75 are more likely to have a flexible employment relationship [10] and this is associated with financial insecurity, but self-employed workers reported retiring at a later age and more motivated to work longer compared to permanent employees [11]. It may be that the higher degree of autonomy that is part of self-employment increases the motivation to continue working [11]. Many employees will have to be replaced in the coming years because they change profession or stop working. If the incentive to work longer is reinforced, the participation rate of older workers will increase and the demand for replacement will decrease [10].

If both employers and employees value the continued participation of older workers in the workforce, then organizations need to understand how jobs can be designed to help

older workers continue to work successfully. When changes take place in an organization, the ways different departments operate are affected, and work or task design is a way to mitigate these changes. Job design is about creating and changing the content, structure, and environment of jobs and roles in organizations [12]. The specifications of the job must meet the requirements of the organization as well as the person working in the job. It involves systematic organization of tasks and responsibilities into a unit of work to achieve the company's performance needs along with individual employee skills, needs and motivation.

Job design is a core part of human resources management and an essential tool to ensure high job satisfaction among employees within an organization and thereby improve overall productivity. A well-designed job can make employees more productive and satisfied. Job design and its particular work characteristics affect a variety of important employee, team, and employer outcomes [5,12], such as employee motivation, job satisfaction, and work performance.

### 1.2. Job Design Theories

There are several theories about how job design can contribute to work outcomes like employee motivation and job satisfaction. The two-factor theory [13] states that the presence of intrinsic motivational factors of the job, such as recognition, challenge and responsibility can increase job satisfaction, while the absence of extrinsic hygiene factors, like pay and working conditions can lead to job dissatisfaction. As a result, subsequent job design focused on increasing the intrinsic motivational factors in a job to improve job satisfaction.

The job characteristics theory [14] also argues that motivation, job satisfaction, and employee performance can be increased through the enrichment of five intrinsic job characteristics: autonomy, task identity, task significance, skill variety, and job-based feedback. This is an indirect relationship in which these job characteristics initially lead to three psychological states of employees feeling responsible, knowing the results of their work, and finding their work meaningful. These in turn lead to internal motivation to perform better.

A somewhat different approach is taken by the job demand–control model (JDC) [15] which does not look at specific job characteristics, but shows that jobs with high demand, such as time pressure, heavy workload and role ambiguity, cause stress that can be balanced with a high degree of control, including work skills, relationships with colleagues and autonomy, which can lead to the psychological wellbeing of employees.

Likewise, the job demands–resources model (JD-R) [16] states that certain job characteristics can be divided into job demands and resources that, in the right balance, also influence employee outcomes [17], such as job satisfaction [18]. According to the JD-R model, resources, such as autonomy, role-clarity, supervisor coaching, and career opportunities, can buffer the negative impact of demands, like work pressure [16,19] and can therefore motivate employees [17]. Some job demands can be differentiated as job challenges, for example workload, time pressure, job complexity, and responsibility, which are obstacles that may be overcome and may elicit positive effects to employees' development, satisfaction, and work engagement [20]. This can help explain why when the demands and resources are high in a job there is more task enjoyment and commitment. Specific job characteristics, such as the amount of task variety and autonomy, and the opportunity for social interaction in a job, also contribute to the internal motivation of employees [21]. An increase in these characteristics should lead to more job satisfaction and better performance. As the decision to continue working is related to motivation and job satisfaction, this work outcome may also be facilitated by job design.

### 1.3. Motivational, Social, and Contextual Job Characteristics

The job design theories have been brought together in the Work Design Questionnaire (WDQ), a comprehensive measure of job design [5], with 21 work characteristics divided into three categories. Motivational, social, and contextual work characteristics

have a demonstrated impact on various behavioral, attitudinal and wellbeing-related work outcomes [22].

Motivational work characteristics reflect the task and knowledge requirements of the job, including autonomy, variety of tasks, skills variety, and job feedback [5] that can motivate employees. As older employees have extensive work experience and crystallized intelligence, they may be able to work well autonomously [23]. Autonomy affects both job satisfaction and performance [22] and can contribute to successful aging at work. The Selective Optimization and Compensation (SOC) theory [24] is a lifespan theory describing adaptation strategies for successful aging, that can be linked to the work context to investigate the relationship between age and job characteristics [25], as certain job characteristics could support employees adapt to age-related changes in work. The SOC theory would explain that autonomy can contribute to successful aging at work. More autonomy should therefore lead to more satisfaction and the improved performance of older workers [26]. While some variety in job tasks is likely to be attractive to most employees [21], the SOC theory would suggest that older workers are less interested in job aspects related to job expansion, meaning more tasks, such as performing more different tasks [27] or a wide variety of tasks [4]. Job complexity [25], related to the difficulty of tasks, can be more attractive to older workers because it allows them to use their accumulated knowledge and skills to perform challenging work. Task significance is how one's job affects the lives and work of others, leading to perceived meaning [14]. Years of experience and knowledge provide older workers with a range of skills and the SOC theory would argue that this variety of skills gives them the satisfaction of applying their accumulated work knowledge and satisfying the desire to pass it on to others (generativity) [28]. It can lead to higher job satisfaction if organizations give older workers the opportunity to use and share their skills [25]. By increasing job complexity, through provision of decision making and knowledge sharing opportunities, employees can expect to have work-related opportunities in the future [25], which may influence their expectation of working longer.

Social work characteristics consider the interpersonal and social aspects of work, including coworker and supervisor support [5]. Social support is about how many opportunities for advice and help from others there are within a job, especially from the supervisor, or from colleagues, and opportunities for building friendships [29]. Within the framework of the job demands–resources model [16], social support from supervisors is a social energy source at work that has a positive influence on the work engagement of employees [30]. Support from supervisors as well as from colleagues is recognized as having a strong impact on employees' wellbeing [31]. In contrast, older employees who experience little social support from colleagues or are even excluded by colleagues are more likely to retire early from working life [32]. Supervisor support can cushion the impact of stress [33] and can also affect an employee's attitude toward work and intention to leave the organization [34]. According to the Socioemotional Selectivity (SES) theory [35], a life-span theory of motivation, older adults focus on their social networks to meet their emotional needs. As people get older, they realize that they have limited time to spend on achieving social goals, and thus they are more likely to focus on the social characteristics of their job. The SES theory would predict that older workers are more focused on meaning in their work than on the acquisition of professional skills or career opportunities [23]. Improving psychosocial working conditions, which are interpersonal and social interactions that influence behavior and development in the workplace, may help keep older workers in employment [36,37].

Contextual work characteristics relate to the physical and environmental contexts in which the work is performed, including working hours, equipment use, and ergonomics [5] or are seen as job demands that relate to surrounding conditions, such as noise [38]. These characteristics describe the physical requirements of the job, which are known to be affected by the age of the employee [26]. Flexible work conditions have been highlighted in studies as a solution to keep older workers in the workforce and are consistently identified as one of the most effective strategies for attracting and retaining older workers [39]. Flexible work-

ing hours are considered facilitators to a longer working life beyond retirement age [40]. Older workers who want to participate in the labor force past the traditional retirement age prefer to find work in an adaptable workplace which offers flexible work schedules, flexible number of hours worked, flexible work locations, flexibility in exiting and re-entering the workforce, and flexible change of jobs and careers [41]. Hence, employers who do actively prepare for the aging workforce still tend to focus on workplace conditions such as these flexible work options [41]. However, contextual characteristics that are focused on easing the work tend to have little effect on job satisfaction and the desire to work longer, and they do not appear to actually contribute to the participation and employability of older workers [42].

### 1.4. Older Workers in Different Job Types

To support an extended working life, it is desirable for work to meet the needs and capabilities of older workers and therefore organizations should design work so that employees can stay in work longer instead of trying to change workers to adapt to work [43]. That is why it is important to fit the job to the older employee in order to encourage working longer. There are a limited number of empirical studies investigating relationships between age and job design [26], but in order to satisfy employees of all ages at work, it is necessary to examine which job characteristics suit them best [4].

As the group of older employees is large and diverse, it is important to gain a better understanding of the differences between older employees to promote effective job design. A limitation of some job design studies is that the sample was heterogeneous in terms of job types, but that job type was controlled in the analysis [4]. More specifically, there has not been much research on individual differences between jobholders that moderate the impact of work characteristics on work outcomes [26]. However, expected differences in work characteristics between occupations were already detected by the WDQ. For example, autonomy is higher in jobs in professional occupations than in jobs in nonprofessional occupations [5]. Certain work characteristics may be more available in some job types than in others. As such, diversity in job types probably results in greater variability in work characteristics. It is impossible for organizations to take individual needs into account and design jobs for each employee [44], but it can be efficient to tailor a set of work characteristics for each type of job.

Job classification is a scheme for clustering jobs based on objective aspects, such as responsibilities and tasks, in order to study jobs in a holistic perspective. Job classification can be used to analyze the similarities and differences among jobs [45] and to compare jobs across organizations. The study sample incorporated three types of job roles: Operational, Professional, and Managerial, selected from the Equal Employment Opportunity Commission (EEOC) job categories [46]. Employees in operational roles perform duties that support day-to-day operations of the company and their focus is on administration, facilities or logistics. Operational jobs are usually based in the office and include positions such as secretaries, receptionists, office managers, administrative staff, and other general office workers. Professionals are knowledge workers who are highly knowledgeable or skilled in a particular area with an emphasis on delivering and applying expertise. Jobs in the professional category include teachers, nurses, engineers, accountants, pilots, lawyers, and programmers. Employees in managerial positions deploy resources and personnel to accomplish work as well and provide direction to their subordinates. Such jobs include directors, middle managers, team leaders, and supervisors.

Although research already argues that job design should consider preferences and motivation of older workers [5,26,44], little is known about how job design can contribute to additional work outcomes, such as motivation to continue working. Systematic review shows limited examples of studies that associate work characteristics for older workers with their willingness to continue working [38]. Work characteristics most often examined are social support and job control that have a positive relationship with the motivation to continue working [38]. Work times, such as flexible working hours, were also often investi-

gated, but no conclusive relationship was found with the motivation to work longer [38]. Learning opportunities have been extensively researched with evidence that it has a positive effect on the ability to continue working (employability) but not on the willingness to work longer [38].

Many European studies apply a quantitative cross-sectional design and involve a sample of respondents from wider economic sectors in a particular country to present the results. For example, job autonomy was positively related to willingness to continue working for older workers (50–65 year) from various industries in the Netherlands [47]. Some studies zoom in on a particular profession, such as Dutch construction workers who are less able to continue working when job autonomy and supervisor support are low [48]. Low autonomy has also been associated with early retirement of Danish nurses [49] and organizational support is positively related to the motivation to continue working in the same job of Dutch high school teachers [50].

To date, limited research has been conducted into whether job roles might impact the relationship between work characteristics and willingness to continue working [38]. This qualitative research takes an alternative approach by subdividing work characteristics into job types, so that it becomes known whether older employees in different jobs assess certain work characteristics equally or differently. Subsequently, quantitative research can be used to investigate whether the differences found between the job types are significant and thus provide input for an effective design for older employees so that they are able and willing to continue working.

This study contributes to the literature on job design and older workers' motivation to continue working. Moreover, extant research does not differentiate between different job types [4], despite strong signals that a good person–job fit [51] between the job and personal factors like gender or financial pressure [37] are strongly associated with work outcomes of older workers. By distinguishing between different types of jobs, this research contributes to both job design and older worker research by unraveling how different types of jobs should be designed to motivate older workers to keep working. In addition, the study results can be useful in practice, by educating managers and HR officers about the characteristics of sustainable work design for older workers performing different types of jobs, and by assisting older workers craft their jobs as they age to encourage them to continue working.

This qualitative study uses a job-diverse sample and aims to investigate how job type influences work characteristics older workers need to continue working and to identify possible differences between operational, professional and managerial jobs. Specifically, three research questions will be answered: (1) How do older workers appreciate work characteristics in their current job? (2) Which work characteristics do older employees need to work longer? and (3) How do these work characteristics differ among older workers in three job types: operational, professional, and managerial roles?

## 2. Materials and Methods

The aim of the present study is to investigate how job type (operational, professional and managerial jobs) influences work characteristics older workers need to continue working.

### 2.1. Sample

Older workers (55+) working in educational and healthcare jobs in the Netherlands participated in this qualitative study. The aging of the population is clearly visible in the healthcare sector, where the largest share of 55+ employees is employed [52]. A large part of teaching and nursing staff will retire in the near future so there is a great need to replace departing staff, but it appears that far too few young people are following an education to be able to replace them all [10]. In addition, health organizations not only have to deal with employees who retire, they also have to deal with a growing demand for services since retired elderly will eventually rely on healthcare. Due to the anticipated and

potential structural staff shortages, it is particularly important for both sectors that their older employees are encouraged to continue working and it is relevant for organizations to look at ways in which the employability of their older employees can be increased.

The U.S. Equal Employment Opportunity Commission (EEOC) classifies all jobs into nine comprehensive job categories with definitions and examples, including Supervisors, Professionals, Technicians, Sales Workers, Administrative Support Workers, Craft Workers, Operatives, Laborers and Helpers, and Service Workers [46]. This job classification is intended to help employers determine whether there is overlapping responsibility between roles and to allow for job comparisons, but can also be used, for example, for equal performance appraisals and salary, recruitment and training purposes [46]. This study uses the EEOC classification scheme in a simpler form to divide the participants into three broad job groups based on the type of work they perform: (1) Operational, similar to EEOC Administrative Support Workers; (2) Professional, similar to EEOC Professionals; and (3) Managerial, similar to EEOC Supervisors. Estimates of the U.S. Department of Labor [53] predict that the highest turnover will occur in these three job types and it is therefore important to keep employees in these jobs longer.

In this exploratory and explanatory study, nonprobability sampling techniques were used to gain more insight into the under-researched population of older employees in certain job types. Two health and two educational organizations were approached with an introduction letter containing information about the study and after approval from HR directors, employees were given an invitation to participate. Nonprobability, purposive, homogeneous sampling [54] was used to recruit a minimum of five participants from each job type based on the following selection criteria: age between 55 and 65 years old (maximum of 10 years before retirement to make the study relevant for participants); permanent employment contract; employed three years or more by the same employer; working in one of the three job types. Participants were included according to their willingness to participate, and they were scheduled for a one hour interview. Ethical considerations were applied by informing participants of the voluntary nature of participation, the option to withdraw from the study at any time, and ensuring their confidentiality.

Regarding demographics, in total 21 participants were interviewed with an average age of 61 years (youngest 56, oldest 64). The majority was female ($n = 13$) and married and their partners were still working ($n = 13$). The level of education attained was high school or lower vocational education ($n = 2$), high vocational ($n = 11$) or university education ($n = 8$). With regard to health status, the vast majority indicated their health was "excellent" or "good" ($n = 17$) and a minority reported their health was "moderate" or "poor" ($n = 4$). All participants had a permanent employment contract and worked three years or more for the same employer in their current job. When considering the job type, more than a quarter of the sample consisted of participants working in managerial jobs ($n = 6$), almost half of the participating employees performed work in professional jobs ($n = 10$), and a quarter of the participants worked in operational roles ($n = 5$).

### 2.2. Method

This qualitative study examined the experiences of older employees, from their own perspective, about their current and possible future job within the context of their current work [55]. This explorative method was applied to clarify the research questions using manifest and latent content analysis [56] and was considered suitable for obtaining credible data on the work experiences of older employees, because such insights can be crucial to appropriate job design for the older workforce.

Semistructured interviews were conducted to collect data from older workers on what they perceive as important work characteristics in their current job and what work characteristics they think they need to continue working. The primarily open-ended questions, to encourage free expression of views and to leave room for new topics, are based on an extensive review of work design and work characteristics [5,22,26]. Typical questions were as follows: (1) "Can you briefly describe your current job?" (2) "What

were good and difficult work experiences in the past week? Why?" (3) "Which elements are important in your current job? (4) "What does working longer mean to you?" and (5) "What elements of your job would make you want to work longer?" Additionally, questions 3 and 5 were supported by a checklist of 21 work characteristics from the WDQ [5] that allowed the interviewer to ask about a particular work characteristic that respondents had not previously addressed in their answers to the open questions.

The interviews were conducted by the first author, and the interview script was tested in a trial interview with two older workers not included in the study to increase clarity of terminology. To ensure rigor in the data collection process, the interviewer took notes, audio-recorded all interviews, transcribed each interview using the actual words, and summarized job information for each question. The transcripts were reviewed by the participants, and their feedback comments on minor inaccuracies were included in the final version of the interview report. It was anticipated that this approach would enrich the understanding of work experiences of older workers and what they need to continue working rather than retire.

*2.3. Analysis*

Using the matrix approach [57], a full thematic sentence-by-sentence analysis of the interview transcripts was performed. Emerging themes were identified, interpreted, and coded into a priori work characteristics [5]. They were categorized into motivational, social and contextual work characteristics, then tallied, and compared to identify patterns between the operational, professional, and managerial job types of older workers.

The full transcript of the interview was read to obtain a general perspective and after that each sentence was coded to identify the topic, which was then interpreted and coded as an emerging theme. The authors discussed these themes, compared them with those of similar content in other transcripts, and decided on the appropriate coding of the a priori work characteristic. Finally, the identified work characteristics were subdivided into three main categories of work characteristics [5]. An example illustrates this content analysis approach. In answer to the question "Which elements are important in your current job?" the respondent replied: "I like to get compliments from my boss when I do something right" (topic). This quote was interpreted as "Appreciation from management is important" (emerging theme) which was coded as "Supervisor support" (a priori work characteristic) belonging to the category "Social work characteristics". The transcripts were reread to ensure that coding of work characteristics matched the content of the interviewees' responses.

Analyzed data was used to discover meaningful patterns between three job types: operational, professional, and managerial staff, to understand similarities and differences between the roles. From 21 transcripts of interviews, a total of 210 work statements were collected that older workers used to describe their work experiences and future job expectations. The statements were coded into 14 work characteristics and then divided into three categories: motivational characteristics 44% (92 out of 210 quotes), social characteristics 34% (72 out of 210), and contextual characteristics 22% (46 out of 210 quotes).

## 3. Results

Older employees participating in this study painted a clear and coherent picture of a range of work characteristics they believe are valuable in their current job and needed to continue working. The most frequently mentioned work characteristics are related to work content, social relationships at work, and working hours. Motivational work characteristics arise from the task environment and are related to the nature and level of work and the manner in which these tasks are to be performed [5]. Participants reported six characteristics, including autonomy, use of expertise and knowledge, focus on key tasks and task adjustments, challenges in work content, personal development opportunities, and psychological workload. Social work characteristics arise from the social environment, referring to work relationships and the way in which employers and employees interact

and perform their duties together [5]. Four items of social characteristics are reported by participants, including cooperation with colleagues and teamwork, mentoring role, support from supervisor, and interaction with clients. Contextual work characteristics that arise from the organizational environment are about the work space and working conditions [5]. Participants reported four characteristics, including flexible working hours, comfortable workspace, use of new technology, and job aids to relieve their physical workload.

### 3.1. Work Characteristics Per Job Type
### 3.1.1. Operational Job Type

As Table 1 column 2 shows, older employees in operational positions most often reported the work characteristics in ranking order of clients, autonomy, colleagues, supervisor, work hours, expertise, and workspace as important in their current job. In order to keep working longer, column 3 shows that this ranking has shifted slightly to include task adjustments, workspace moved up the ranking, and colleagues, work hours, and expertise are not required to continue working.

**Table 1.** Operational job type profile of older workers with important work characteristics in current and future jobs.

| Operational Jobs | | | |
|---|---|---|---|
| **Work Characteristic** | **Current Job** | **Future Job** | **Description** |
| **Motivational** | 2.Autonomy | 2.Autonomy | Work independently; not having anything imposed upon them; not being continuously checked. |
| | - | 3.Tasks | Keep core tasks; meaningful work; tasks taken over by backup; no radical changes in the range of duties. |
| | 6.Expertise | - | Be point of contact for others for general knowledge; apply practical and life experience. |
| **Social** | 1.Clients | 1.Clients | Help people and please them; representative work for clients; customer friendliness; deliver quality. |
| | 3.Colleagues | - | Social contact; sharing personal things; supporting each other; getting assistance; team spirit; solidarity. |
| | 4.Supervisor | 5.Supervisor | Receive appreciation; open communications and discussion; sympathetic relationship. |
| **Contextual** | 5.Work hours | - | More hours but fewer days; later start times; flexible holidays. |
| | 7.Workspace | 4.Workspace | Comfortable furniture; quiet workplace with minimal noise; limited number of locations; some time outside. |

Number = ranking, based on the importance of the work characteristic as reported by the respondent: 1 = highest ranking, 7 = lowest ranking.

Older workers in operational roles highly value the interaction with clients in their current job (ranking 1) and also clearly indicated that they need this work characteristic to keep working (ranking 1). They focus on having general conversations and providing a good service, rather than giving specific advice to clients.

> *"I want to do representative work for clients; helping people and pleasing them, making them happy." (O5) "Because of their life experience, older employees have more sense of service provision to customers."* (O2)

Older employees in operational job types also highly valued autonomy in their current job (ranking 2). To continue working, they reported that they should at least maintain their current level of autonomy (ranking 2), with clearly defined personal responsibilities, while looking to reduce responsibility to others.

> *"I think it is important that I can set my own priorities. I want to determine what is important now and what needs to be done quickly. Basically I want to decide what my day looks like." (O4) "As an older employee I don't want anything imposed on me. It is nice if you can work independently." (O2) "I want clearly defined tasks."* (O3)

Overall, operational staff is satisfied with their key tasks in their current job. To continue working, they reported that they wanted to keep their core tasks (ranking 3). They would appreciate some help with tasks that can be done by someone else, such as a younger colleague.

> *"If my responsibilities change radically, my job satisfaction would be gone."* (O3) *"I want to keep doing the same tasks, because I can keep it up."* (O6)

> *"It is more fun when certain tasks are done by younger colleagues."* (O1)

> *"I could use some backup, someone who can take over some tasks."* (O3)

Older employees in operational positions expressed appreciation for a quiet workspace with comfortable furniture in their current job (ranking 7) and the need for this to work longer (ranking 4).

> *"Ideally, I would like a fairly quiet workplace in the office, alternated with spending some work time away from the office."* (O4) *"I am not a fan of open-plan offices."* (O3) *"I would like to work at one location, instead of two."* (O2) *"I don't want to be just indoors, but also like to be outside."* (O1)

In their current job, operational staff indicated they appreciate getting support from their supervisor (ranking 4), in particular receiving appreciation and open communication, and they believe that a good relationship with their supervisor is essential to be able to work longer (ranking 5).

> *"Support from my manager becomes increasingly more important as I get older. I especially expect clear open communication with mutual discussions.* (O2) *"I must have a sympathetic relationship with my supervisor. If that's not right, I wonder what reason I would have to keep working."* (O5)

Older workers in operational job types cited collaboration with colleagues, flexible work hours, and use of expertise as important elements in their current job, but did not see it as a condition for continuing to work.

### 3.1.2. Professional Job Type

As Table 2 column 2 shows, older employees in professional positions most often reported the work characteristics in ranking order of work hours, clients, expertise, mentoring, colleagues, supervisor, autonomy, job aids, and workload as important in their current job. In order to keep working longer, column 3 shows that this ranking has shifted slightly to include task adjustments, and colleagues, supervisor, autonomy, and job aids are not required to continue working.

Older workers in professional roles highly appreciate flexible working hours in their current job (ranking 1) and future job (ranking 1) and clearly indicated that they need this work characteristic to continue working. They value flexibility in both the number of working hours and the start/end times. To continue working, older workers in professional roles expressed the need for reduction of working hours to spend some time at home for hobbies, volunteer or care activities, flexible start and finish times, more ad hoc days off and longer holidays to recover from the workload.

> *"I want to be flexible and do not mind working an extra hour, but I would like to start later on another day."* (P3) *"I think a good daily routine is important."* (P7) *"Flexibility works both ways, sometimes I work a little less so that I can recharge, and if necessary I work an extra day."* (P10) *"I no longer want to work 60 h, there must be a balance between work and personal life."* (P8)

**Table 2.** Professional job type profile of older workers with important work characteristics in current and future jobs.

| Work Characteristic | Current Job | Future Job | Description |
|---|---|---|---|
| **Professional Jobs** | | | |
| **Motivational** | - | 3.Tasks | Continue doing the same tasks; mix of thinking and execution tasks; fewer admin tasks; more ancillary/advisory tasks; special projects. |
| | 3.Expertise | 5.Expertise | Recognized for substantive work; use of experience; innovative, creative; informal/structured ways of knowledge and skills sharing. |
| | 7.Autonomy | - | Set own priorities; certain degree of freedom; accountability for work outcomes; own responsibility. |
| | 9.Workload | - | Reasonable psychological workload; realistic goals and less time pressure; no work to take home. |
| **Social** | 2.Clients | 2.Clients | Intensive client contact; helping, supporting; focus on customer target group; more time with individual clients. |
| | 4.Mentoring | 4.Mentoring | Teaching, advising, so others can do it themselves; help people discover; share practical stories; grooming young people. |
| | 5.Colleagues | - | Working in teams; working with younger people who have the latest informative and knowledge; exchanging ideas. |
| | 6.Supervisor | - | Good contact with manager; involvement in management decisions |
| **Contextual** | 1.Work hours | 1.Work hours | Fewer hours; later start and end times; fewer weekend/night shifts; schedule own hours; longer holidays; work–life balance. |
| | 8.Job aids | - | Tools to relieve physical activities; massage chair to reduce stress. |

Number = ranking, based on the importance of the work characteristic as reported by the respondent: 1 = highest ranking, 9 = lowest ranking.

Professionals frequently mentioned that they value interaction with clients in their current job (ranking 2) and future job (ranking 2). They indicated that they enjoy intensive client contact and that they want to help clients with their expertise.

*"I will have to keep in touch with customers otherwise I will not be able to work."* (P1) *"I want to give support and help others, so that people can call on me."* (P6) *"I now feel calmer in my interaction with clients due to my increased knowledge and broader experience in dealing with people."* (P5)

Although professionals are satisfied with their key tasks in their current job, they also welcome some task adjustments to continue working (ranking 3), such as a shift to more ancillary and advisory tasks or an emphasis on specific initiatives and projects and less on administrative tasks.

*"I want to continue to perform my core tasks."* (P3) *"If I have extra tasks that I enjoy, the work will also be easier for me."* (P1) *"I would like to get rid of some regulation, registration, and administrative tasks."* (P9) *"I would like to do more consultancy work to use my brain more."* (P10)

Older workers in professional positions reported the importance of mentoring others in their current job (ranking 4) and also stressed the need for mentoring opportunities in a future job (ranking 4). They would appreciate informal as well as more structured ways of coaching colleagues.

*"I would like to help colleagues with my knowledge and experience, I teach them something and advise them, so they can do it themselves."* (P5) *"In addition to my normal work, I would like to mentor interns. There is nothing more fun than that."* (P7) *"I firmly believe in the student–apprentice–master principle. Therefore, the last phase of my career will be successful if I can help with grooming younger staff."* (P6) *"I think it is important to look beyond the boundaries of my role, to take on additional tasks, such as supervising interns, and then I will be able to last longer in my job."* (P4)

Professionals indicated that they highly valued having expertise in their current job (ranking 3), meaning a depth of knowledge in a particular area. To continue working, professionals indicated the need for knowledge transfer and use of their experience (ranking

5). Most professionals are very keen on informal or structured opportunities for sharing their knowledge and experience with colleagues in their organization.

> *"I think that large service organizations have a need for knowledge transfer."* (P6) *"I would like people to use my knowledge, but it is their choice whether they do so."* (P10) *"Listening to my experience is important to me, not to be proved right, but to have the discussion."* (P9) *"I will have to continue to use my knowledge and experience; otherwise I will not be able to work."* (P1)

Older workers in professional job types cited support from colleagues and supervisor, autonomy, use of job aids, realistic workload as important elements in their current job, but gave it a lower ranking compared to the other work characteristics and did not see it as a condition for continuing to work.

### 3.1.3. Managerial Job Type

As Table 3 column 2 shows, older employees in managerial positions most often reported the work characteristics in ranking order of mentoring, autonomy, colleagues, challenges, work hours, personal development, and technology as important in their current job. In order to be able to work longer, column 3 shows that the top of this ranking remains almost the same, with task adjustments being added and challenges, work hours, and technology are not required to continue working.

**Table 3.** Managerial job type profile of older workers with important work characteristics in current and future jobs.

| **Managerial Jobs** | | | |
|---|---|---|---|
| **Work Characteristic** | **Current Job** | **Future Job** | **Description** |
| **Motivational** | 2.Autonomy | 3.Autonomy | Being own boss; freedom; accountable; plan and prioritize; take decisions; to be able to say "no." |
| | 4.Challenges | - | Mentally demanding tasks; creativity, problem solving; new ideas; variety; work in a modern way. |
| | - | 4.Tasks | Phasing out certain core tasks; letting go; contribute; perform problem analysis; interim work; organizing chaos; focus on goals. |
| | 6.Personal development | 5.Personal development | Gather new ideas; inspiration; self-study; get a boost; reading books, attending seminars, courses outside of work. |
| **Social** | 1.Mentoring | 1.Mentoring | Energize, enthuse people; teaching; people development; give advice; discover talent. |
| | 3.Colleagues | 2.Colleagues | Bring people together for a common cause; facilitate processes between people; help others do better; teamwork. |
| **Contextual** | 5.Work hours | - | Later start times; longer periods off work; more time to relax. |
| | 7.Technology | - | Keep up with IT developments; use range of software and gadgets. |

Number = ranking, based on the importance of the work characteristic as reported by the respondent: 1 = highest ranking, 7 = lowest ranking.

Older workers in managerial job types highly valued mentoring in their current job (ranking 1) and also clearly indicated that they need this work characteristic to keep working (ranking 1). They define the mentoring role as coaching junior colleagues, assisting colleagues, and supervising trainees.

> *"I enjoy discovering talent and to develop younger colleagues."* (M1) *"I want to energize people and be a good teacher."* (M4) *"I like to explain things to my colleagues so they get excited and know what to do."* (M5)

Older employees in managerial positions also frequently reported to value collaboration with colleagues in their current job (ranking 3) and future job (ranking 2). They enjoy bringing colleagues together, facilitation of discussions, and helping others.

> *"Working with colleagues is important, we're in it together, we depend on each other, need each other, and together we can determine if things are going well."* (M2) *"I want*

*to connect people, do it together, then you can make progress."* (M6) *"I want to facilitate a team where people have a lot of expertise."* (M4)

Managers also indicated the importance of autonomy in their current job (ranking 2). To continue working, managers reported that they should at least maintain their current level of autonomy (ranking 3) for their own decision-making and work planning to be able to continue working.

*"Autonomy is important for everyone who works: it is the degree of control over what you do, your own freedom and responsibilities."* (M1) *"I need freedom, to be autonomous, to plan and prioritize my own work."* (M6)

*"I want freedom to think about what I want to achieve."* (M2)

Managers are satisfied with their core tasks in their current job. To continue working, they reported that they wanted to keep their core tasks (ranking 4). Some managers mentioned that they are eager to complete their tasks with pride and focus on delivering high quality.

*"I want to focus on tasks that have a visible result, that lead to a clear improvement that is important for the organization."* (M6) *"I like to keep my core tasks, finish the work, and deliver quality."* (M4) *"I would like to perform analyses to solve problems or conduct an interim project."* (M1)

Only older employees in managerial positions expressed appreciation for personal development opportunities to expand their knowledge and progress in their current work (ranking 6) and to work longer (ranking 5). Continuous learning for managers means gathering new ideas and inspiration.

*"I want to continue developing myself, to obtain ideas, and get inspiration to keep going; I need that boost."* (M5) *"Participate in a think-tank so that I get a lot of inspiration and perspectives."* (M1) *"Sometimes I need to be alone to read a book, to get new ideas."* (M6)

Older workers in managerial job types cited having challenges, flexible work hours and use of new technology as important elements in their current job, but gave them a lower ranking compared to the other work characteristics and did not see them as a condition for continuing to work.

*3.2. Similarities and Differences between Job Types*

Table 4 shows there are similarities and differences between older workers performing in the three job types: operational, professional and managerial. When comparing these, it shows that managers gave the highest ranks in motivational characteristics of all three groups, professionals ranked higher than the other groups in contextual characteristics, and operational staff has the highest ranking in social characteristics of all groups.

In their motivational work characteristics, all three job types indicated that they wanted to focus on their core tasks and that they need task adjustments in their future job. In addition, job types have a number of work characteristics in common. Older workers in operational and managerial roles reported the need for keeping their levels of autonomy to be able to continue working. In their social work relationships, older workers in professional and managerial job types emphasized the importance of mentoring when working longer. Furthermore, older workers in operational and professional positions both value interaction with clients to continue working.

There are a number of job characteristics that are unique to a job type that distinguish it from the other two job types. Unique work characteristics for operational roles are supervisor support and comfortable workspace. Professionals especially want to use their expertise and flexible working hours, and managers are different because they value personal development and collaboration with colleagues.

**Table 4.** Ranking of work characteristics that older employees, per job type, consider important in their current and future job.

| Work Characteristic | Current Job | | | Future Job | | |
|---|---|---|---|---|---|---|
| | O | P | M | O | P | M |
| **Motivational** | | | | | | |
| Autonomy | 2 | 7 | 2 | 2 | | 3 |
| Expertise | 6 | 3 | | | 5 * | |
| Tasks | | | | 3 | 3 | 4 |
| Challenges | | | 4 * | | | |
| Personal development | | | 6 * | | | 5 * |
| Workload | | 9 * | | | | |
| **Social** | | | | | | |
| Colleagues | 3 | 5 | 3 | | | 2 * |
| Mentoring | | 4 | 1 | | 4 | 1 |
| Supervisor | 4 | 6 | | 5 * | | |
| Clients | 1 | 2 | | 1 | 2 | |
| **Contextual** | | | | | | |
| Work hours | 5 | 1 | 5 | | 1 * | |
| Workspace | 7 * | | | 4 * | | |
| Technology | | | 7 * | | | |
| Job aids | | 8 * | | | | |

O = operational job type; P = professional job type; M = managerial job type. The rank number is based on the frequency with which a work characteristic is reported by the respondents: 1 = highest ranking, 9 = lowest ranking; * = unique to job role.

## 4. Discussion

This qualitative study empirically explored work characteristics that are appreciated by older workers in their current jobs and that are needed to work longer to be able to provide directions for designing sustainable jobs for older workers. An increase in the labor participation of older workers is desirable because the age structure of the working population is changing, and structural labor shortages are expected [1]. This study focused on older (55+) workers in three job types and on work design as a possible and appropriate intervention to support working longer. In general, the study results show that older employees gave the highest rankings to social characteristics, then motivational characteristics, and then contextual characteristics. Therefore, the findings expand knowledge of these work characteristics and support the relevance of social work characteristics [58] in relation to suitable work design for older workers.

Social characteristics reflect the fact that work is performed within a broader social environment and are viewed by employees as a major aspect of work [59]. Creating high-quality relationships in the workplace is important [60]. Levels of support from supervisor and colleagues are both related to turnover intention; however, supervisor support is more strongly related to positive work attitudes [34]. Both social characteristics are also quoted frequently by the participants. Improving social support for older workers shows desirable effects on work ability and work motivation [58].

The presence of motivational characteristics is expected to enrich the job [61]. Older workers in this study indicated that motivational characteristics are important and therefore increasing motivational characteristics would serve to increase job satisfaction [62] and possibly also the intention to work longer. Positive psychosocial [37] work factors, such as a high degree of influence, recognition from management, and opportunities for development increase the chance of working longer [63].

Contextual characteristics include the terms of employment and the physical work environment. Many scholars as well as policy makers focus on flexible working options [37,41,64] as a major intervention to support older workers in extending their working life. The results of this interview study also support this because the quotes on contextual

characteristics, in particular flexible working hours, suggest that these work characteristics need to be continuously included in the design of jobs for older workers.

Some work characteristics were considered important in the current job, such as challenges, workload, technology, and job aids, but were not considered necessary in a future position. By contrast, tasks related to work content were not much discussed as part of the current job but were considered important in a future job. As shown in Figure 1, all three job types indicated they want to focus on their core tasks and specific task adjustments to be able to continue working.

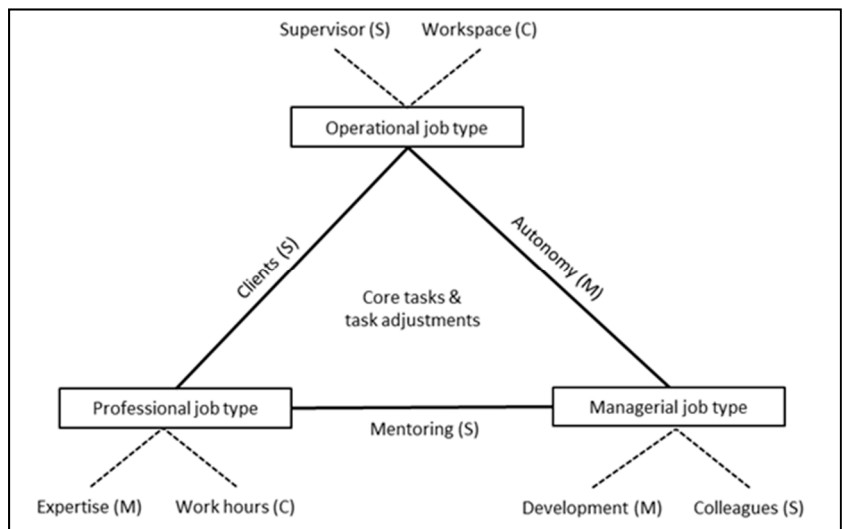

**Figure 1.** Similarities and differences of work characteristics between job types.

### 4.1. Operational Job Types

Older employees in operational roles described work characteristics they need to work longer (see Figure 1) as interaction with clients (in common with professional staff), autonomy (in common with managers), but also reported support from supervisor and comfortable workspace (both unique to this job type).

Operational roles are generally jobs that involve duties that need to be fulfilled for others, hence the importance of working with internal clients. They share this work characteristic with staff in professional roles that provide expert services to internal and external customers, but they interpret it at a different level. Operational employees focus on having more general conversations with internal customers, such as their colleagues. Autonomy remains important in the future job, because an operational person needs autonomy to work independently, in forward-thinking and problem-solving ways.

Interestingly, there is a shift in social characteristics, where collaboration with colleagues was important in the current job, the support of the supervisor becomes more important to work longer. Social support may be a more important resource in the operational job type. Operational employees are under internal pressure because of the direct evaluation of services by external clients and colleagues [65], and social support from the supervisor can be an important resource [66], as it can motivate these employees to maintain a positive attitude towards clients [67]. Supervisor support can be seen as a more predictable and higher form of support than support from colleagues because providing consistent instrumental and emotional support is often seen as an important part of supervisors' work and they also have the skill and experience to do so [34]. Various studies have shown that when people have a good relationship with their boss, they are more motivated, try harder and perform better. This can be explained by the fact that their work is often directly related to their relationship with their manager, having to divide their time between business and personal tasks as assigned during the workday [68]. Supervisor

support is defined by employees as receiving rewards and appreciation [34], increasing the likelihood of working beyond retirement [47,69].

In order to be able to work longer, the workplace and ergonomics become important [5]. Operational employees are the only ones that highlighted the importance of a comfortable work environment, with a low ranking in the current job, which increases sharply as the employee describes what it takes to work longer. Older employees in operational roles perform their work most of the time in the office. Due to the support function to other employees, the physical presence of the operational person in the office is appreciated, making the comfort of the workplace more important as the employee gets older.

### 4.2. Professional Job Types

Older employees in professional roles described work characteristics they need to work longer (see Figure 1) as interaction with clients (in common with operational staff), mentoring (in common with managers), but also reported work hours and using their expertise (both unique to this job type).

Professionals are characterized as knowledge workers who have a high level of expertise, education, and experience, and focus on creating, applying, or disseminating knowledge [70]. Organizations rely on professionals, especially older professionals with their crystalized intelligence, for efficient problem-solving and the provision of tailored services, and hence the company aims to retain them [71]. Interaction outside the organization, such as interacting with clients, is higher in professional jobs [5], because these workers are specifically focused on providing expert services to others. Professionals generate or use knowledge to improve products, processes or services and as they get older are driven to share and transfer knowledge and skills to their younger colleagues [72], either by "use of expertise" or "mentoring" younger colleagues. The work of professionals emphasizes interaction with customers, the use of skills from previous experiences, and the ability to pass on knowledge to younger workers. All of these contribute to successful aging [73].

For workers in professional roles, good supportive coworker relationships reduce intention to leave [74]. Rather than furthering their own careers, older workers may prefer mentoring others, which is consistent with SOC theory [24]. They appreciate mentoring colleagues to continue working. This links to the use of expertise since structured mentoring can be a vehicle for knowledge transfer. An older worker can stay active by mentoring and encouraging others to increase their knowledge. Participants indicated that knowledge can flow in both directions, as they also want to learn from the mentee about new ideas and trends [75].

Professionals are the only group to mention work pressure in their current job, although they convert that into the need for flexible working hours for a future job. In general, older adults focus more on positive experiences and activities that provide emotional satisfaction than on knowledge acquisition [5]. The JD-R model adopts the active learning hypothesis [15] to explain that high demand/high control jobs challenge employees to develop new skills. Therefore, high demand (workload) and high resources (support from colleagues) can thus lead to skill development for professionals, which is an important outcome for them, reflected in their need to use their expertise in a future job.

The work often happens in the head of the knowledge worker, and this thinking process can take place in the office as well as outside it [70]. Today's knowledge workers are characterized by their flexible working style [70] so offering flexible working hours and working conditions seem to appeal to this job type. Flexibility is no longer a hygiene factor, but an important condition for older professionals to work longer.

In general, professionals like a lot of autonomy at work and do not want their manager to tell them what to do or control them [70], which can explain why they score low on supervisor support and do not see this work characteristic as necessary for working longer. Professionals do not necessarily need more autonomy when working longer, but certainly they do not value a reduction. Limited autonomy can lead to demotivation and reduced desire to work longer. Their need for autonomy can also mean a trade-off with flexibility.

Flexible working conditions offer a lot of autonomy, and autonomy is needed to work flexibly. Autonomy can be a mediator because if professionals already have a lot of autonomy, they will be able to handle flexible working conditions to do their work well. If they can work flexibly, the direct control of their manager will be reduced, their autonomy will be greater and with that their motivation will be higher. Thus, they will be more likely to work longer.

### 4.3. Managerial Job Types

Older employees in managerial roles described work characteristics they need to work longer (see Figure 1) as mentoring (in common with professionals), autonomy (in common with operational staff), but also reported collaboration with colleagues and personal development opportunities (both unique to this job type). Interestingly, studies find moderate evidence that challenging work has a positive relationship with motivation to continue working [38]. However, in this study, managers have indicated that they value challenges in their current job, but they do not necessarily need them to work longer.

Autonomy is often present in positions with managerial or supervisorial tasks and responsibilities [62]. Managers are typically given many different assignments, requiring them to know the big picture as well as details in order to draw conclusions and make decisions [76]. That is why they must develop sufficient capabilities and therefore opportunities for development are important for them to be able to do a good job in the future. Personal development opportunities are often included as a job resource [77]. In addition, a manager's job can have a high workload with many different ill-defined nonroutine activities and frequent interruptions, working under pressure, often after hours, and leaving little time to think [78]. Therefore, development opportunities might offer a way to stand back, reflect, and learn. Work-related learning can be seen as an outcome of the motivational process [79] because learning can be stimulated by job resources such as autonomy [15]. Autonomy is higher in managerial jobs [5], since these jobs typically involve complex, nonroutine work where higher levels of self-direction are present. Compared to other job types, managers enjoy a high degree of autonomy and freedom to make choices and decisions [80]. As this is inherent to the job, the manager will want to maintain that sense of freedom in order to continue working.

In addition to their own development opportunities for growth and progress, managers also value the development of others. An important part of their job is that managers need to direct, respond to, energize and engage others [76]. Therefore, most of their time is spent in the company of other people: around 10% of time is spent on communication with superiors, and most time is allocated to subordinates [80], which they also define as social relationships with colleagues. Support from colleagues is a very important work characteristic of successful aging, and it has been recommended to design jobs for older workers that offer the opportunity to collaborate with others [81]. Support from colleagues, for example from other managers, can reduce personal stress by sharing similar experiences and showing sympathy [34].

As managers are responsible for the development of their staff, mentoring is naturally seen as an important work characteristic for future work. Mentor behavior of managers is expressed in, for example, challenging assignments, coaching and sponsoring of the mentee [82], while older professionals in the role of mentor focus more on knowledge and skills transfer and the development of the mentee's expertise. Employees who work beyond retirement want to mentor their younger counterparts [69]. In order to retain employees, careful career guidance by the manager can contribute as part of an aging policy [83]. Managers who provide others, especially their direct reports, with career-related mentorship are more likely to be promoted and thereby help themselves advance their careers [82]. Therefore, when managers focus on developmental relationships, and act in the role of mentor, not only does the mentee benefit, but it is also a personal development opportunity for the manager.

### 4.4. Limitations and Future Research

This study provides new insights into the expectations of older workers in different job types about how certain work characteristics can support them to work longer, but the findings may be limited by a number of factors. First, the specific context of health and educational organizations in which the participants were examined may limit the generalizability of the findings. Employees in other industries may give a different appreciation to the work characteristics for working longer. Second, using interviews as a research method was beneficial for obtaining depth and richness of older workers' experiences of work and expectations towards working longer. However, the findings from this study are drawn from a relatively small sample of older employees who all work in some type of job, namely performing operational, professional, or managerial tasks. This limits the applicability of the findings to a wider group of 'white-collar workers' or to older people working in jobs that require manual labor. In addition, given that research participation was voluntary, the sample was likely biased toward older workers who are interested in the challenges of work and aging. Finally, the results of this study cannot confirm or explain correlations between work characteristics, nor between work characteristics and the outcome of working longer. Therefore, we propose to apply the Work Design Questionnaire [5] to older workers in future quantitative research, as a measure to examine the relationships between job design and working longer. This would make a useful contribution to the development of job designs for older workers to promote continued participation in their current job.

### 4.5. Implications for Theory and Practice

This study has several implications for theory. The interview results showed which work characteristics are valued as important by older employees for continuing to work and thus contribute to the scarce literature that integrates aging, work motivation and job design [26].

First, in line with Morgeson and Humphrey [5], this study supported the idea that older workers like other groups within the workforce, recognize and classify work characteristics into three main categories: motivational, social and contextual work characteristics and that these are instrumental to the impact of job design on the outcome of working longer. Therefore, the WDQ with additional work characteristics can be used as a measure to identify work characteristics that fit each job type.

Second, we argue that the motivation to continue working depends on certain work characteristics, which differ per job type. A limitation with research on job design for older workers is that work characteristics, that could improve work outcomes such as job satisfaction and willingness to continue working, are controlled in the analysis [4] and generalized across occupations, assuming that individuals will experience the same work characteristics in different jobs [5].

Finally, the primary contribution of this study is job type profiles that present which distinctive work characteristics are considered important to older workers in operational, professional and managerial jobs to stay in employment. This study found that work characteristics are not perceived in the same way by all older workers, which can influence their retirement decisions. As far as is known, no other study has empirically distinguished between job types for older workers. Therefore, the findings of this research support literature that recognizes the importance of understanding the role of individual differences in examining the effects of work characteristics on work outcomes [5]. The results also contribute to job crafting theories [84], an individualized process that allows older workers to personalize the task, social and contextual aspects of their job with the aim of improving their work experience [85].

Simultaneously, this study also has several implications for practice. The results of this interview study are intended to inform managers and human resources practitioners about the characteristics of sustainable work design for older workers who are performing in different job types. As the number of older employees has increased, organizations find it difficult to account for the individual needs of each employee [44], and it is beneficial to



be able to design jobs for different job types to keep employees working longer. Therefore, instead of individual contracts to facilitate working longer, companies can make agreements with groups of employees in the same job type.

In addition, older employees who wish to remain in paid employment are provided with an overview of work characteristics that may help them in developing a suitable job for the future. If older workers in different types of jobs require different work characteristics, the results of this study can assist them in crafting their jobs as they age and adjust the characteristics to support successful ageing at work [84]. In this way, organizations can support job crafting by older workers who have the interest and understanding to design their jobs to meet their future needs.

## 5. Conclusions

The aim of this qualitative study was to investigate how job type influences work characteristics older (55+) employees need to continue working and to identify possible differences between operational, professional and managerial jobs. This was a timely study conducted to contribute to the knowledge of suitable job design for older workers to enable them to extend their working life. This research focused on questions of what older workers consider important in their current job, what job characteristics they need to continue working, as well as how and why there are differences between the types of jobs.

Most respondents reported the importance of staying focused on core tasks and getting task adjustments, but there are also notable differences between the job types. Older workers in operational roles need work characteristics including interaction with clients, autonomy, comfortable workspace, and supervisor support. Older workers in professional roles need flexible working hours, interaction with clients, mentoring roles, and use of expertise. Older workers in managerial roles need mentoring roles, collaboration with colleagues, autonomy, and personal development opportunities.

In conclusion, the results show that certain work characteristics have a different impact on the design of future jobs for older workers, depending on the type of job of the employee. The current study thus adds more insight into which work characteristics can benefit older workers to influence the work outcome of working longer.

**Author Contributions:** H.d.B. and T.v.V. developed the conceptual research design. The interviews and data analysis were carried out by H.d.B. with the support of T.v.V. and J.d.J. H.d.B. wrote the first version of the manuscript and the later versions were edited together by all three authors. All authors have read and approved the submitted version.

**Funding:** This research received no external funding.

**Data Availability Statement:** The data presented in this study are available on request from the corresponding author.

**Acknowledgments:** We are very grateful to all participants for taking part in this study. Furthermore, we would like to thank Edward Groenland of the Nyenrode Business University for his advice and concrete assistance with regard to the analysis of qualitative research data. Finally, we would like to thank Jacqueline Truesdale for her editorial changes to our manuscript.

**Conflicts of Interest:** The authors declare no conflict of interest.

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
