# Peer review of "Job Design to Extend Working Time: Work Characteristics to Enable Sustainable Employment of Older Employees in Different Job Types"

_sustainability, doi:10.3390/su13094719_

Round 1
Reviewer 1 Report
Thank you for the opportunity to review the manuscript, "Job Design to Extend Working Time: Work Characteristics to Enable Sustainable Employment of Older Employees in Different Job Types," for Sustainability. I enjoyed reading your work.
I have only a few suggestions for the manuscript, and offer these comments in the spirit of trying to help you polish the article into the best form it can take. I hope that they are useful.
First, I wondered about the referencing sequence. I've seen manuscripts where the numbered citations appear in order (1, 2, 3, and so on) with the reference list corresponding to those points. I've also seen alphabetized lists, where the in-text citations correspond to those numbers (e.g., the first citation might be to 35, rather than 1). But neither of these seem to be the case here, and it might be good to check the citation structure with the journal guidelines.
Somewhere, perhaps on page 2, it might be useful to incorporate a discussion of the increasing statutory retirement age (still 66 in 2021 in the Netherlands), and the consequences of working past that age: e.g., loss of unemployment and disability benefits, requirement for a new contract. While there are forces driving employees to work longer, there are also barriers and obstacles - beyond job design - that constrain those choices.
I would also like to see a discussion (possibly p. 6?) of why you chose health and education as sectors for study. I can imagine that they have a wide range of jobs, spanning operational, professional, and managerial types, but you could also have looked at governmental employment, business and commerce, IT, or transportation. Why the ones you selected?
I also caught a few small infelicities that you might want to correct:
- page 2, lines 53-62: It seems like the reasons for #1 (help companies avoid talent shortages and knowledge loss) and #2 (keep a company’s knowledge and networks) are repetitious, or at least overlapping to a great degree. How is "avoiding knowledge loss" and "keeping knowledge" meaningfully distinguishable?
- page 5, line 224: The category "programmers" is listed twice.
- page 6, lines 260-264: There is something tautological going on here: "According to the U.S. Equal Employment Opportunity Commission’s job classification guide [34], there are three job categories or job types that define the type of work performed: 1) Operational, 2) Professional, 3) Management and according to estimates of the US Department of Labor [35], the expected highest turnover will occur across these three job types." Yes, I suppose if there are three forms of employment, the highest turnover would occur within these three forms.
You might want to check the English language with a grammar check, or ask a native speaker to give it a look. For the most part, the English is excellent, but there were a few turns of phrase that seemed out of place (e.g., top line of Table 1: "not being imposed on anything" rather than "not having anything imposed upon them").
It's a nice piece of work - the sample is modest but the contribution is real. It does indicate where there are overlapping and divergent desires for continuing workers.
Author Response
We have provided our responses to Reviewer 1 comments in the attached Word document.

Reviewer 2 Report
Thank you for allowing me to review this research. I read this study and thought about many things in my job and it helped me a lot in my studies. I don't think it needs to be modified in particular, but I leave a few comments to say that it would be better if it was supplemented a little bit.
This study should highlight in the introduction the need for this study and the implications that this study can bring.
What are some of the existing studies in Europe? And what is the difference between this study and the existing one? It should be presented.
Author Response
We have provided our responses to Reviewer 2 comments in the attached Word document

Reviewer 3 Report
The paper is pertinent and relevant, reporting a well designed and well described research. A theoretical framing is provided, presenting theories relevant to the research under analysis and justifying its pertinence and adequacy. The authors base this analysis on classical as well as recent production in the field.
Participants and sampling are well described, and ethical concerns were taken into account. When detailing the methods, there is reference to a work characteristics checklist (pp. 6-7). It would be useful to present that checklist to the reader.
When presenting the analysis it is not clear what the unit of analysis was (word, paragraph, sentence, idea) and whether codes were emergent or a priori. This may be further detailed in the analysis.
Data are presented in a clear fashion – tables were appreciated – and illustrated by quotes from the participants. Tables might improve in terms of readability if horizontal lines were kept and shading was used to convey intensity (if possible).
There is a clear analysis of how the data relates to previous theory, as well as to practical implications. The authors also recognize the limitations of their study.
Although I am not a native English speaker myself, I have noted (in yellow on the document in annex) a few expressions, which may benefit from revision. In particular, calling participants units of analysis does not seem adequate (p. 6).

Author Response
We have provided our responses to Reviewer 3 comments in the attached Word document.
